

# Identification of ecogeographical gaps in the Spanish *Aegilops* collections with potential tolerance to drought and salinity

Rosa María Garcia[1], Mauricio Parra-Quijano[2] and Jose María Iriondo[3]

[1] Centro Nacional de Recursos Fitogenéticos (CRF-INIA), Alcalá de Henares, Madrid, Spain
[2] Facultad de Ciencias Agrarias, Universidad Nacional de Colombia, Bogotá D.C., Colombia
[3] Área de Biodiversidad y Conservación, Universidad Rey Juan Carlos, Móstoles, Madrid, Spain

## ABSTRACT

Drought, one of the most important abiotic stress factors limiting biomass, significantly reduces crop productivity. Salinization also affects the productivity of both irrigated and rain-fed wheat crops. Species of genus *Aegilops* can be considered crop wild relatives (CWR) of wheat and have been widely used as gene sources in wheat breeding, especially in providing resistance to pests and diseases. Five species (*Ae. biuncialis*, *Ae. geniculata*, *Ae. neglecta*, *Ae. triuncialis* and *Ae. ventricosa*) are included in the Spanish National Inventory of CWRs. This study aimed to identify ecogeographic gaps in the Spanish Network on Plant Genetic Resources for Food and Agriculture (PGRFA) with potential tolerance to drought and salinity. Data on the Spanish populations of the target species collected and conserved in genebanks of the Spanish Network on PGRFA and data on other population occurrences in Spain were compiled and assessed for their geo-referencing quality. The records with the best geo-referencing quality values were used to identify the ecogeographical variables that might be important for *Aegilops* distribution in Spain. These variables were then used to produce ecogeographic land characterization maps for each species, allowing us to identify populations from low and non-represented ecogeographical categories in *ex situ* collections. Predictive characterization strategy was used to identify 45 *Aegilops* populations in these ecogeographical gaps with potential tolerance to drought and salinity conditions. Further efforts are being made to collect and evaluate these populations.

Corresponding author
Rosa María Garcia,
rosamaria.garcia@inia.es

## INTRODUCTION

Drought is one of the most important abiotic stress factors limiting biomass (*Araus et al., 2002*), and, consequently, it significantly reduces crop productivity (*Lambers, Chapin & Pons, 2008*; *Noorka & Pat Heslop-Harrison, 2014*). Wheat (*Triticum* spp.), the second most-produced cereal in the world (*FAO, 2013a*), can be severely affected by this type of abiotic stress. For instance, in 2009, wheat yield in Kenya dropped by 45% due to drought, compared to 2010 production that took place under a good crop season. Australia, which

suffered multi-year droughts between 2002 and 2010, experienced a 46% drop in wheat yield in 2006 (*FAO, 2013b*).

Another threat to both irrigated and rain-fed wheat crops is salinization (*Mujeeb-Kazi & Diaz de Leon, 2002*). Salt stress and drought have similar effects on structural and functional aspects of plants (*Al-maskri et al., 2014*), but salt stress also causes ion toxicity and ionic imbalance (*Hameed, Ashraf & Naz, 2011*). Drought stress decreases photosynthetic efficiency, greatly reducing growth and development (*Al-maskri et al., 2014*). Salt-affected soils occur in all continents and under almost all climatic conditions. However, they are more widely distributed in arid and semi-arid regions than in humid regions (*Abrol, Yadav & Massoud, 1988*). Many crops in these areas are grown under irrigation, but inadequate irrigation management may lead to secondary salinization (*Glick et al., 2007*). Large areas of naturally saline and alkaline soils account for 6% of the world's land surface. These saline soils have never been cultivated because present major crops are salt-sensitive (*Fita et al., 2015*).

The species of genus *Aegilops* have been widely used as gene sources in wheat breeding, especially in providing resistance to pests and diseases such as leaf, stem and stripe rusts (*Puccinia recondita*, *P. graminis* and *P. stiiformis*) or hessian fly (*Mayetiola destructor*). Numerous studies have searched for drought and salt stress tolerant genotypes. For instance, *Xing et al. (1993)* studied the potential of some *Aegilops* species, including *Ae. ventricosa* Tausch, as gene donors in breeding for salt tolerance. Subsequent studies proposed *Ae. ovata* L. as a source of salt tolerance in wheat (*Farroq, 2002*) and determined that *Ae. ovata* and *Ae. biuncialis* Vis. have wide genetic variation for salt tolerance (*Colmer, Flowers & Munns, 2006*). *Mólnar et al. (2004)* compared the physiological and morphological responses to water stress in *Ae. biuncialis* and *Triticum aestivum* L. genotypes, and concluded that *Ae. biuncialis* genotypes from dry habitats have greater drought tolerance than wheat, making them good candidates for improving drought tolerance in this crop. *Mondini, Nachit & Pagnotta (2015)* identified SNPs variants conferring salt tolerance in durum wheat. Due to their present and potential use as gene donors in wheat breeding, *Aegilops* species can be considered crop wild relatives (CWR) of wheat (*Heywood et al., 2007*). Five of these species (*Ae. biuncialis*, *Ae. geniculata* Roth, *Ae. neglecta* Req. ex Bertol., *Ae. triuncialis* L. and *Ae. ventricosa*) are included in the Spanish National Inventory of CWRs (*Rubio Teso et al., 2013*).

Ecogeographical land characterization (ELC) map can be helpful in determining different adaptive scenarios of a species in a given territory. An ELC map represents the different ecogeographical conditions in which a particular species or group of species occurs, using some variables of high importance in the species' distribution likely to be determinant for the adaptive landscape (*Parra-Quijano, Iriondo & Torres, 2012a*).

*Parra-Quijano et al. (2008)* developed an ELC map for Peninsular Spain and the Balearic Islands, using different sources of ecogeographical information. The ability of the ELC map to discriminate different areas with different adaptive pressures was tested with eight crop and CWR species. They found that the ELC map had an effective discriminatory capacity to delineate adaptive scenarios. Because the efficacy in detecting plant adaptation was heterogeneous for the species analyzed, it was concluded that the application of

ecogeographical maps for detecting plant adaptation may be better approached through an ad-hoc ELC map generation for each target species. The ELC concept is detailed in depth in *Parra-Quijano, Iriondo & Torres (2012a)*, and several applications related to the collection, conservation and efficient use of plant genetic resources have been developed (e.g., *Parra-Quijano, Iriondo & Torres, 2012a*; *Parra-Quijano, Iriondo & Torres, 2012b*; *Thormann et al., 2016*).

Gap analysis has been widely applied for conservation purposes (*Maxted et al., 2008*). For instance, *Ramírez-Villegas et al. (2010)* applied a gap analysis methodology to collect the crop genepool of *Phaseolus* beans and evaluated conservation deficiencies at three different levels (taxonomic, geographic and environmental). *Khoury, Laliberté & Guarino (2010)* reviewed global crop and regional conservation strategies and recognized the importance of filling gaps in genebanks of plant genetic resources. Recently, *Shehadeh, Amri & Maxted (2013)* carried out a gap analysis of *Lathyrus* L. species. In this study, predictive distribution maps for each *Lathyrus* taxon were produced based on climatic data, and *ex situ* conservation gaps were identified as regions where the species was predicted to occur but seed accessions had not been previously collected, or, alternatively, the species was under-sampled.

Optimized Collecting Design (OCD) is a technique described by *Parra-Quijano, Iriondo & Torres (2012b)* that involves the identification of ecogeographical gaps for a target species in a target *ex situ* genebank. Based on the premise that genetic adaptation is achieved through natural selection acting upon particular limiting environmental conditions, this technique aims to improve the genetic representativeness of genebank accessions by improving their ecogeographical representativeness. Like other gap analysis techniques, OCD compares the collecting locations of the target species accessions currently held in the genebanks and the species' occurrence data from external sources (spatial gaps). It then uses ELC maps to detect adaptive scenarios not represented in the target *ex situ* genebank. Using ELC maps to design collecting strategies can help to include accessions from marginal or under-represented environments that may contain important traits related to adaptations to biotic and abiotic stress.

Another interesting issue for genebank managers and stakeholders is the identification of genotypes that can have a specific use in plant breeding. The Focused Identification of Germplasm Strategy (FIGS) is a useful approach for screening large germplasm collections to identify sets of accessions with a high probability of containing specific target traits based on the ecogeographical information of the sites where the populations were collected (*Mackay & Street, 2004*). If we know where a set of *ex situ* accessions has evolved, or at least where they have grown for a period long enough for adapted genotypes to have been selected, we can establish relations or patterns between the environmental conditions of the site and the presence or absence of the target trait. We can then make predictions on non-evaluated germplasm (*Mackay & Street, 2004*).

In recent years, FIGS has been successfully used to identify sources of resistance to sunn pest in wheat in Syria (*El Bouhssini et al., 2009*) and to Russian wheat aphid in bread wheat (*El Bouhssini et al., 2011*). FIGS has also been used to identify traits related to abiotic stresses, such as drought adaptation in *Vicia faba* L. (*Khazaei et al., 2013*). These studies
selected accessions from an ecogeographically-characterized collection that complied with certain values or ranges for the characterized variables, set by the researchers based on their knowledge of the species. Other FIGS approaches can be used to identify accessions of potential interest, when the trait under consideration cannot be directly related to an ecogeographical variable. In this case, partial evaluation data from the target collection are required to detect a reliable relationship between the ecogeographical variables and the trait of interest. This approach has been successfully applied by *Thormann et al. (2016)* and *Endresen et al. (2012)*, who identified sources of resistance to stem rust in bread and durum wheat, and by *Bari et al. (2012)* and *Bari et al. (2014)*, who predicted resistance to stem rust and stripe rust in accessions of wheat landraces.

A broader concept has recently been developed to identify germplasm with a high probability of containing specific target traits, named "predictive characterization". This term comprises a set of approaches that use geographic and environmental data to search for particular traits in a usually large set of populations, not only accessions and landraces but also CWR (*Thormann et al., 2016*). This search can be carried out by means of the ecogeographical filtering method or the calibration method (*Thormann et al., 2016*).

Using these ecogeographical approaches, we aimed to apply OCD and predictive characterization techniques to the optimization of the *ex situ* collection of crop wild relatives of wheat in the national reference genebank of Spain. The objectives of this study were: (1) to identify spatial and ecogeographical gaps within Spain land which could be present in the Spanish genebank collections of *Aegilops biuncialis, Ae. geniculata, Ae. neglecta, Ae. triuncialis* and *Ae. ventricosa* to design an optimized systematic collection strategy of crop wild relatives of wheat for the national genebank of Spain; and (2) to identify non-collected populations that might be of potential interest because of their tolerance to drought and salinity within the group of ecogeographical gaps for each species in order to prioritize their collection.

## MATERIALS & METHODS

### Target species

The targeted species were *Ae. biuncialis* Vis., *Ae. geniculata* Roth, *Ae. neglecta* Req. ex Bertol., *Ae. triuncialis* L. y *Ae. ventricosa* Tausch. Species names were standardized using *Van Slageren (1994)*. Regarding their distribution in Spain, all of them are found in the Mediterranean biogeographical region. *Ae. biuncialis* is also found in the Macaronesian region, whereas *Ae. geniculata* can also be found in the Atlantic region. Table 1 provides a summary of information on distribution, autecology and mating system of each species.

### Species datasets

Data on Spanish populations of *Aegilops* spp. collected and conserved in the genebanks of the Spanish Network of Plant Genetic Resources for Food and Agriculture (PGRFA) (hereinafter "accessions") were obtained from the Spanish Inventory of Plant Genetic Resources (available at http://wwwx.inia.es/inventarionacional/, accessed 26 May 2015).

Data on population occurrences from additional sources (hereafter "external sources") were obtained from the Global Biodiversity Information Facility (GBIF;

Peer J

**Table 1** **Autecology and mating system of the targeted *Aegilops* species.** Information obtained from *Van Slageren (1994)* unless otherwise stated.

| | *Ae.biuncialis* | *Ae. geniculata* | *Ae. neglecta* | *Ae. triuncialis* | *Ae. ventricosa* |
|---|---|---|---|---|---|
| General habitat | Dry and disturbed habitats (fallow, roadsides, edges of cultivation). | | | | |
| Specific habitat | Dry, rocky mountain slopes. | Dry, rocky mountain slopes, wastelands. | Stony fields and hill slopes. Marginal habitats where parent rock surfaces and only pockets of the top soil remain. | Wastelands, sandy wadis (dry riverbeds), and dry rocky slopes of hills and mountains. | Sandy wadis, including saline locations, and even marshy riversides. |
| Vegetation | Various forest types (frequently with *Pinus halepensis* and *Quercus* sp.). Also in grasslands, maquis vegetation and steppe. More rarely in river valleys. | Vegetation types include garrigue, maquis, grassland, shrub, woodlands, forests and scrubs (e.g., of *Quercus, Pinus, Juniperus* and *Pistacia*). Steppe, and, more rarely, also humid pastures, dunes and even swamps. | Vegetation types such as, grasslands, stony fields and hillslopes, maquis, garrigue, in forests or scrubs of e.g., *Acacia, Quercus, Ceratonia* and *Pinus*. Occasionally found on river banks and generally more humid habitats. | Vegetation types include garrigue, maquis, grassland, shrub- and woodlands, (open) forests and scrubs, e.g., of *Acacia, Quercus, Pinus, Juniperus*, and of cultivated *Pistacia, Ceratonia siliqua* and *Liquidambar* trees. Also found in the steppe up to the margin of the desert, but, more rarely, also in humid pastures, river terraces. | Grasslands. Also found in scrubs of *Pistacia* and *Juniperus,* oak forests, and in *Poterium-dominated* vegetation. |
| Soil | Variety of bedrock types: mainly limestone but also on schists, shales, basalt, granite, and pillow lavas.<br><br>Soil textures are mainly clay- or sandy loam, or clay, or (rarely) pure sandy soils. | Bedrock is predominantly limestone but shales, pillow lava, silicate, Mediterranean terra rosa, karst, basalt and sandstone are also reported.<br><br>Soil texture also varies widely: often on clay- and sandy loam, clay, and gravel; more rarely on pure sand and highly organic soil such as turf. | The parent rock is mainly limestone, but less frequently also alluvium, basalt, pillow lava, schists, silicates, and sandstone.<br><br>Recorded soil textures include loam, clayloam, sandy loam, and, more rarely, sands and clay. | Bedrock is predominantly limestone and basalt, but shales, pillow lava, silicate, Mediterranean terra rosa, karst, schist, and sandstone are also reported.<br><br>Soil texture varies widely; often on clay- and sandy loam, (sandy) clay, and gravel; more rarely on loss, pure sands, and marly soils. | Predominantly on soils with a limestone bedrock, far less on basalt or sandstone.<br><br>Recorded soil textures include clay- and sandy loams, less often more pure clay or loam. Growth on very poor, stony soils. |
| Climate | Annual rainfall data of 225–800 mm indicate some drought tolerance, but it also occurs in areas with as much as 1,250 mm. | Wide annual rainfall amplitude, varying from less than 100 mm up to 1,100 mm. | Rainfall data vary from 450 to 750 mm, and in some sites it can be as high as 1,400 mm. | Wide annual rainfall amplitude, varying from 125 mm up to 1,400 mm. | Rainfall data vary widely: from less than 100 mm up to 600 mm, but most are from the range 200–350 mm. |
| Mating system | Considered largely autogamous (*Hammer, 1980*) | Considered largely autogamous (*Hammer, 1980*) but mixed mating has been observed (*Arrigo et al., 2011*). | Considered largely autogamous (*Hammer, 1980*) but mixed mating has been observed (*Arrigo et al., 2011*). | Considered largely autogamous (*Hammer, 1980*) but mixed mating has been observed (*Arrigo et al., 2011*). | Considered largely autogamous (*Hammer, 1980*) but it can occasionally hybridize with wheat (*Van Slageren, 1994*). |

available at http://www.gbif.org/, accessed 14 February 2014) and Anthos (available at http://www.anthos.es/, accessed 25 June 2015) databases. Populations conserved in *ex situ* genebanks that do not belong to the Spanish Network on PGRFA were also considered external sources. Accessions missing in the Spanish National Inventory, provided by the Plant Genetic Resources National Centre of the Spanish National Institute for Agricultural and Food Research and Technology (http://wwwx.inia.es/coleccionescrf), were considered external sources as well.

All accessions and external sources without geographic coordinates were removed. We also removed external sources with geographic coordinates expressed in decimal degrees with less than two decimals in both latitude and longitude or without textual description on the occurrence site, and external sources with geographic coordinates expressed in UTM with lower resolution than 1 × 1 km. Passport and presence data were standardized to CAPFITOGEN data formats which is basically the Multi-Crop Passport Descriptor (*FAO/Bioversity, 2012*) plus four additional administrative fields for collecting or presence site description (*Parra-Quijano et al., 2015*).

Georeferencing data of both accessions and external sources were cleared of spatial intraspecific duplicates. We considered that species occurrences less than 1 km apart belonged to the same population, following *Iriondo et al. (2009)*. Therefore, these population occurrences were considered spatial duplicates and only one of them was considered.

Accessions and external sources free of spatial duplicates were subjected to a geo-referencing quality evaluation using GEOQUAL from the CAPFITOGEN toolkit (*Parra-Quijano et al., 2015*). We set the quality threshold in TOTALQUAL100 = 80, so only records with quality values above this threshold were considered in subsequent analyses.

## Selection of ecogeographical variables

Ecogeographical information was extracted for each occurrence site from raster layers with a 30 arc-second resolution and classified into three ecogeographical components: bioclimatic variables (37), edaphic variables (16) and geophysic variables (4) (see Table S1).

The variables that might be the most relevant for each species in each ecogeographical component were then identified using the SelecVar tool from CAPFITOGEN (*Parra-Quijano et al., 2015*). SelecVar extracts information from the ecogeographical variables (layers) to the occurrence sites and assesses the importance of each variable in generating different adaptive scenarios for a species (*Parra-Quijano et al., 2015*). It estimates variable importance according to the random forest classification (RFC) and detects redundant variables through bivariate correlation analysis. The RFC analysis provides a ranking of the most important variables for establishing ecogeographical categories, placing variables with a higher mean decrease in accuracy in the first positions (*Cutler et al., 2007*). Rankings were obtained for each ecogeographical component for each species. Bivariated correlation analysis detected correlated variables in the top fifteen variables of the RFC ranking. Variables with Pearson correlation coefficient >|0.50| and *p*-value <0.05 in the same ecogeographical component were identified and removed.

## Generation of the ELC maps

An ecogeographical land characterizacion map or ELC map is a representation of the different adaptive scenarios of a species (*Parra-Quijano, Iriondo & Torres, 2012a*). In this study, we generated an ad-hoc ELC map for each target taxa assuming that each taxa may respond differently to the environment. Thus, we considered that this approach would provide better results than those obtained from the generation of a single ELC map for all the taxa.

The top three bioclimatic, three edaphic and two geophysic variables in the rankings of variable importance (i.e., the ecogeographical variables suggested as relevant in the species distribution) were considered in generating the ELC map for each species. The variables latitude and longitude were included as two additional geophysic variables to obtain maps with spatially aggregated categories. The ''elbow'' method was used to create the ecogeographical categories. This is a simple system which uses K means as a clustering algorithm where the cut-off point is determined on the basis of the decrease in the sum of the intra-group squares (*Ketchen & Shook, 1996*). The optimal number of categories is reached when the decrease in the intra-group sum of squares in a range of $n$ and $n+1$ groups is less than 50%. ELC maps were generated for each species using the ELCmapas tool of the CAPFITOGEN toolkit (*Parra-Quijano et al., 2015*) and the following parameters: 30-arc-second cell size resolution and 8 clusters as the maximum number of categories allowed per ecogeographical component (bioclimatic, edaphic and geophysic).

## Identification of spatial and ecogeographical gaps and prioritization of occurrence sites for future collections in *Aegilops*

Spatial gaps were identified based on occurrence site coordinates. For each species, locations cited by external sources more than 1 km apart from accessions were considered spatial gaps. An ecogeographical representativeness analysis of the existing germplasm collections was then carried out to identify ecogeographical gaps (ELC map categories not represented in *ex situ* genebanks where the species occurs).

Areas with a high occurrence of external sources (listed as ecogeographical gaps) and a low occurrence of accessions were considered priority collecting sites, as were areas corresponding to the ELC map categories with a low frequency for the species and in the territory. Occurrence data from external sources were then ranked according to their priority of collection based on the frequency of each ELC map category in the study area, the frequency of each species in each ELC map category and the differences between the external sources data set and the accessions data set. Ecogeographical gap identification and prioritization of external sources occurrence data for germplasm collection were performed using the Representa tool of CAPFITOGEN (*Parra-Quijano et al., 2015*). External sources which occurred in ecogeographical categories not represented by the corresponding species in the genebanks of the Spanish Network, i.e., external sources reclassified by the Representa tool within the range from 1 to 4, were considered priority ecogeographical gaps.

### Selection of collection sites for traits of tolerance to drought and salinity

To search for populations with a higher probability of containing phenotypes with a high tolerance to drought and salinity, the external sources considered priority ecogeographical gaps were filtered using the Lang aridity index (AI$_L$) and topsoil salinity. The Lang aridity index was calculated as:

AI$_L$ = Annual precipitation/(Annual mean temperature).

Following the ecogeographical filtering technique of predictive characterization (*Thormann et al., 2016*), we selected populations occurring in sites with AI$_L$ <40. We considered the 20% of the populations with the lowest Lang aridity index and the highest topsoil salinity (*HWS Database , 2012*) values to be the fraction of interest for each of the five target species.

The process followed to reach the objectives of the study is shown in Fig. 1.

## RESULTS

### Germplasm collection sites and presence data

Data pre-processing showed that the most frequently recorded species and those with the widest distribution in Spain were *Ae. geniculata* and *Ae. triuncialis*. The least frequently recorded species was *Ae. biuncialis*. Table 2 shows the number of accessions and occurrence data from external sources for each species before and after clearing spatial duplicates and applying the geo-referencing quality threshold. The percentage of spatial duplicates in the occurrence data of external sources ranged from 3 to 39%, recorded for *Ae. biuncialis* and *Ae. geniculata*, respectively. Applying the geo-referencing quality threshold decreased the number of non-duplicated populations to 6 and 21% in the occurrence data of external sources of *Ae. biuncialis* and *Ae. geniculata*, respectively.

The species records for *Ae. geniculata* (accessions + external sources) remaining after the clearing of spatial duplicates and the geo-referencing quality threshold are shown in Fig. 2. The maps of the other *Aegilops* species are shown in Figs. S1–S4. The populations represented by accessions in the genebanks are not homogeneously distributed in the studied area (see Fig. 2A), nor are they more frequently located in the areas where the presence of these species was reported by external sources. For instance, 66% of the preserved populations of the five target species (226 out of 345 populations) were collected in the autonomous communities of Extremadura and Castilla-La Mancha, whereas only 10% of the external sources (272 out of 2614 populations) are located in these two communities. The populations reported by external sources are abundant in the south of Spain, in the provinces on the eastern coast and in the autonomous community of Navarra.

### Identification of variables of importance

The variables selected for each studied species and ecogeographical component (bioclimatic, geophysic and edaphic variables) are shown in Table 3. The variables isothermality and altitude were selected in four of the five studied species.

Garcia et al. (2017), *PeerJ*, DOI 10.7717/peerj.3494

**Table 2** Number of *Aegilops* germplasm accessions and occurrence records from external sources with geographical coordinates included in the study before and after clearing spatial duplicates and filtering by the geo-referencing quality threshold.

| | *Ae. biuncialis* | | *Ae. geniculata* | | *Ae. neglecta* | | *Ae. triuncialis* | | *Ae. ventricosa* | |
|---|---|---|---|---|---|---|---|---|---|---|
| | Germplasm accesions | Occurrence records from external sources | Germplasm accesions | Occurrence records from external sources | Germplasm accesions | Occurrence records from external sources | Germplasm accesions | Occurrence records from external sources | Germplasm accesions | Occurrence records from external sources |
| Initial number of georeferenced records | 6 | 30 | 144 | 4,850 | 33 | 870 | 191 | 1,674 | 26 | 363 |
| Number of records removed for having low accuracy[*] | 0 (0%) | 13 (43%) | 0 (0%) | 1,292 (27%) | 0 (0%) | 372 (43%) | 0 (0%) | 580 (35%) | 0 (0%) | 99 (27%) |
| Number of records removed for been considered spatial duplicates | 0 (0%) | 1 (3%) | 20 (14%) | 1,879 (39%) | 1 (3%) | 71 (8%) | 12 (6%) | 221 (13%) | 2 (8%) | 63 (17%) |
| Number of non-duplicated records with TOTALQUAL ≤80 | 2 (33%) | 1 (3%) | 5 (3%) | 317 (7%) | 3 (9%) | 73 (8%) | 5 (3%) | 148 (9%) | 5 (19%) | 43 (12%) |
| Number of non-duplicated records with TOTALQUAL>80 | 4 (67%) | 15 (50%) | 119 (83%) | 1,362 (28%) | 29 (88%) | 354 (41%) | 174 (91%) | 725 (43%) | 19 (73%) | 158 (44%) |

Notes.

*Number of records with geographic coordinates expressed in decimal degrees with less than two decimals in both latitude and longitude or without textual information on the occurrence site, plus records with geographic coordinates expressed in UTM with lower resolution than 1 × 1 km.
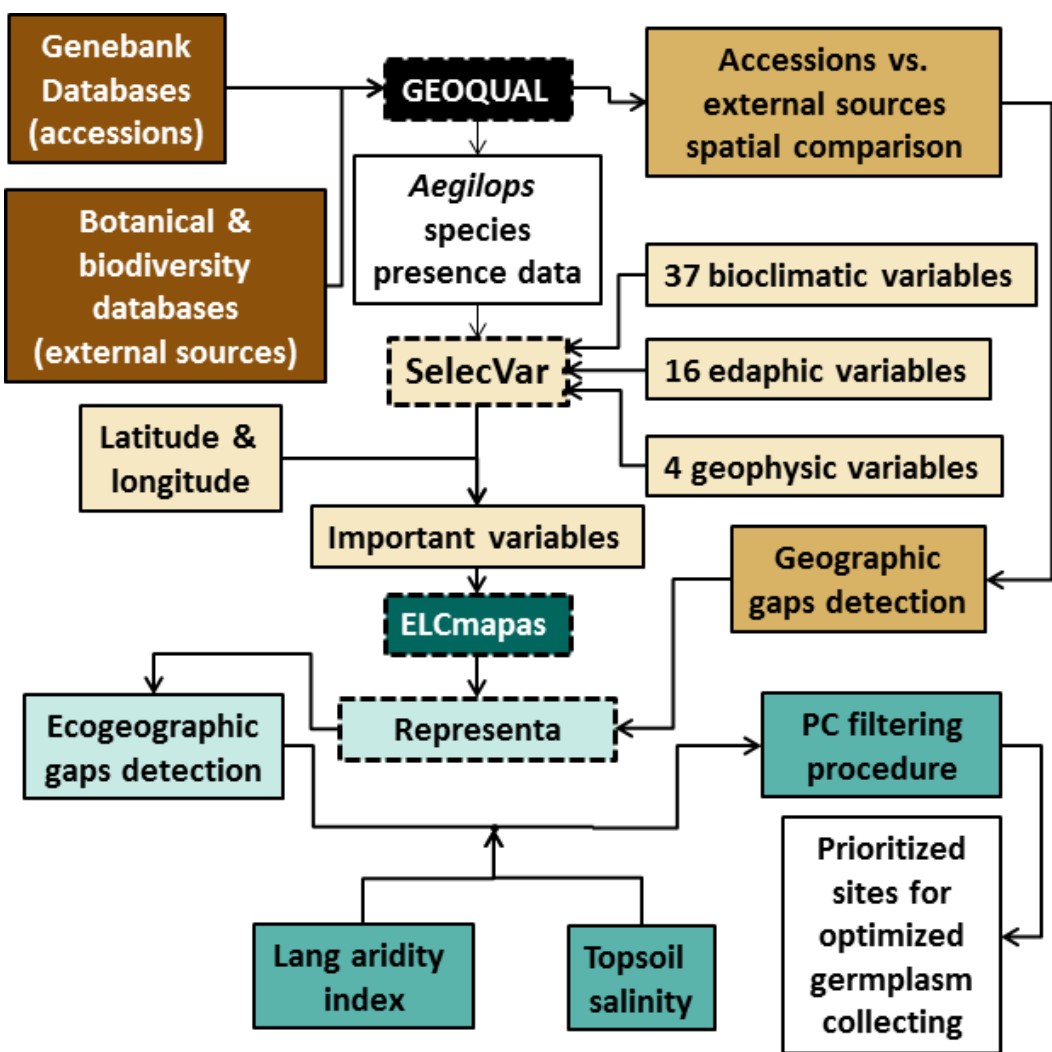

**Figure 1** Process carried out to identify ecogeographical gaps in the Spanish *Aegilops* germplasm collections with potential tolerance to drought and salinity.

## Generation of ELC maps

The ELC map obtained for *Ae. geniculata* is shown in Fig. 3. The maps of the other *Aegilops* species are shown in Figs. S5–S8. The number of ecogeographical categories generated in the maps ranged between 26 (in the ELC map for *Ae. biuncialis*) and 27 (in the maps for the other species). The environmental characteristics of the different categories of each ELC map are summarized in Table S2.

## Identification of spatial and ecogeographical gaps in the *Aegilops* germplasm collections

Among the populations reported by external sources, 2,571 were identified as spatial gaps. *Ae. geniculata* was identified as the species with the largest number of spatial gaps. As seen in Table 4, 393 populations were identified as priority ecogeographical gaps in the registered genebank collections of *Aegilops* in the Spanish National Inventory of Plant
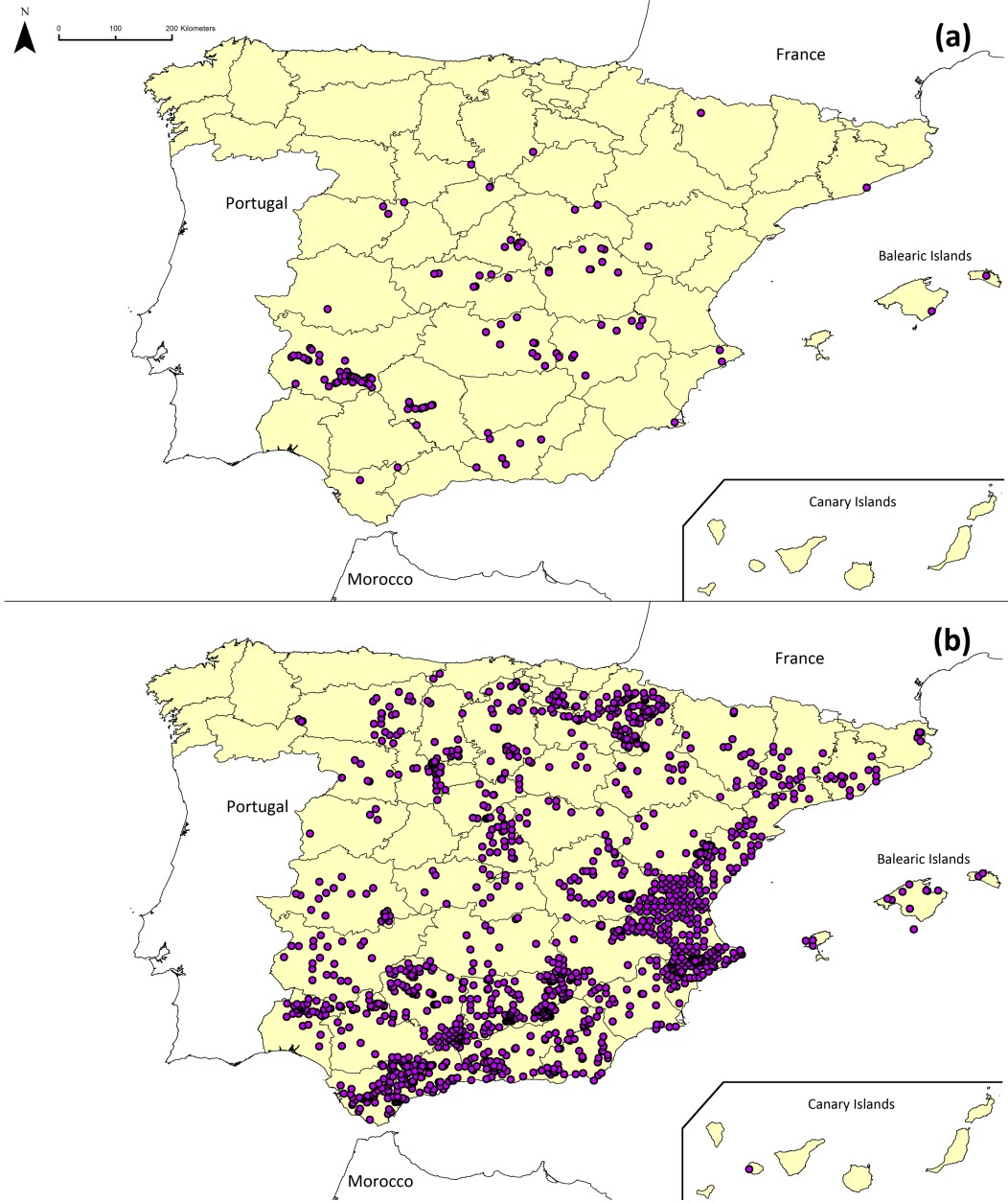

**Figure 2** Location of (A) origin of germplasm accessions in Spain and (B) population occurrence from external sources of *Ae. geniculata.*

Genetic Resources. These 393 populations occur in ecogeographical categories that are not represented by the corresponding species in the Spanish Network. *Ae. geniculata*, the first-ranking species in number of spatial gaps, was also identified as the species with the largest number of ecogeographical gaps. *Ae. biuncialis*, the species with the lowest number of preserved accessions (Table 2), is also the species whose *ex situ* ecogeographical representativeness needs the most improvement, as 80% of the available external sources were identified as high priority gaps. On the contrary, only 10% of the analyzed external

**Table 3 Variables selected in each *Aegilops* species and ecogeographical component according to the 'importance' function of the random forest approach (see Table S1 for variable description).**

| Species | Bioclimatic component | Geophysic component | Edaphic component |
|---|---|---|---|
| *Ae. biuncialis* | 1. January minimum temperature<br>2. Precipitation of the wettest month<br>3. Isothermality | 1. Northness<br>2. Eastness | 1. Topsoil salinity<br>2. Topsoil reference bulk density<br>3. Topsoil base saturation |
| *Ae. geniculata* | 1. Minimum temperature of the coldest month<br>2. Annual temperature range<br>3. Isothermality | 1. Slope<br>2. Altitude | 1. Topsoil gravel content<br>2. Topsoil organic carbon<br>3. Topsoil sand fraction |
| *Ae. neglecta* | 1. March precipitation<br>2. May maximum temperature<br>3. Isothermality | 1. Altitude<br>2. Northness | 1. Topsoil total exchangeable bases<br>2. Reference depth of the soil unit<br>3. Topsoil salinity |
| *Ae. triuncialis* | 1. April minimum temperature<br>2. April precipitation<br>3. Isothermality | 1. Eastness<br>2. Altitude | 1. Topsoil clay fraction<br>2. Topsoil organic carbon<br>3. Topsoil base saturation |
| *Ae. ventricosa* | 1. Mean temperature of the coldest quarter<br>2. April precipitation<br>3. Temperature seasonality | 1. Altitude<br>2. Slope | 1. Topsoil sodicity<br>2. Topsoil reference bulk density<br>3. Topsoil base saturation |

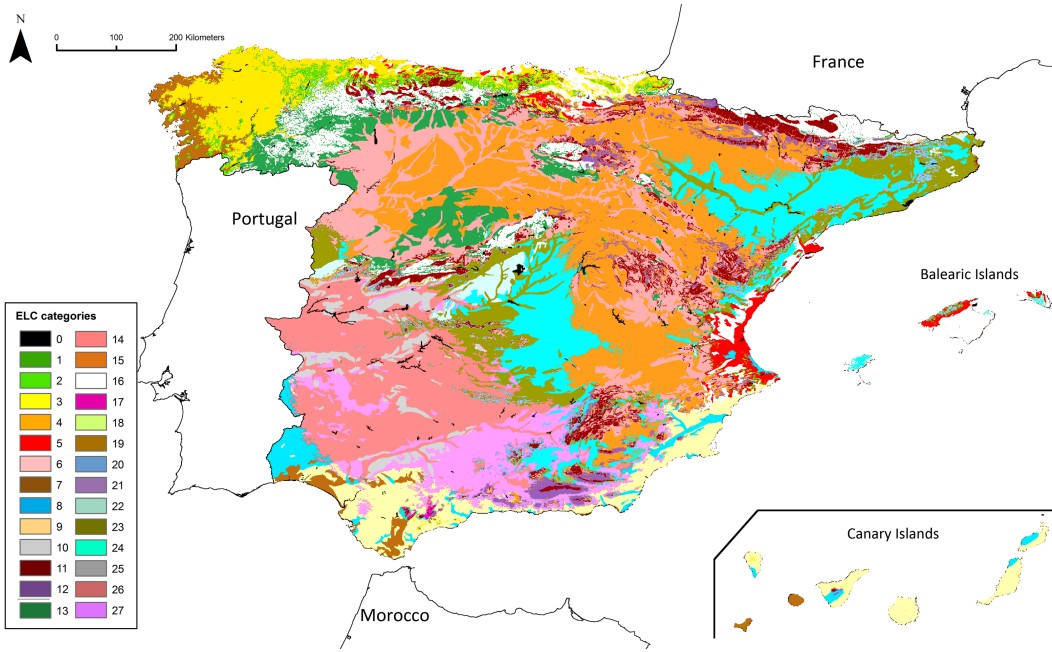

**Figure 3 ELC map of *Ae. geniculata* for Peninsular Spain, the Balearic Islands and the Canary Islands.** The environmental characteristics of the different categories are described in Table S2.

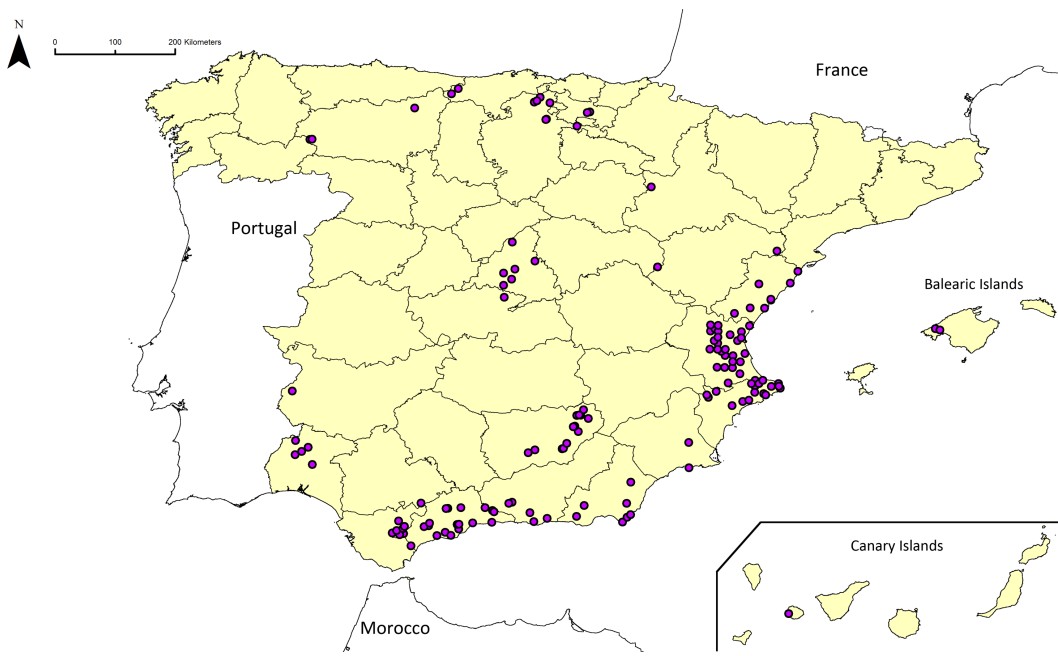

**Figure 4** Location of the *Ae. geniculata* populations identified as priority ecogeographical gaps in Spain.

**Table 4** Number of population occurrences from external sources of *Aegilops* subjected to representativeness analysis and number of spatial gaps and priority ecogeographical gaps identified in Spain.

|  | *Ae. biuncialis* | *Ae. geniculata* | *Ae. neglecta* | *Ae. triuncialis* | *Ae. ventricosa* | TOTAL |
|---|---|---|---|---|---|---|
| Number of population occurrences from external sources | 15 | 1,362 | 354 | 725 | 158 | 2,614 |
| Number of spatial gaps | 15 | 1,359 | 317 | 722 | 158 | 2,571 |
| Number of priority ecogeographical gaps | 12 | 140 | 133 | 73 | 35 | 393 |
| Percentage of population occurrences from external sources identified as priority ecogeographical gaps | 80 | 10 | 38 | 10 | 22 | – |

sources of *Ae. geniculata* and *Ae. triuncialis*, the two species with the highest number of preserved accessions (Table 2), were identified as priority ecogeographical gaps.

The geographic distribution of the populations identified as priority ecogeographical gaps of *Ae. geniculata* is shown in Fig. 4. The maps of the other *Aegilops* species are shown in Figs. S9–S12.

Including germplasm from priority ecogeographical gaps in the genebanks of the Spanish Network on PGRFA would significantly improve their ecogeographical representativeness (number of ecogeographical categories not currently represented in the Spanish Network,

**Table 5** Number of ELC categories for *Aegilops* currently represented in the Spanish Network and potential increase (%) in representativeness after collecting priority ecogeographical gaps.

| | Ae. biuncialis | Ae. geniculata | Ae. neglecta | Ae. triuncialis | Ae. ventricosa |
|---|---|---|---|---|---|
| Number of categories in the ELC map | 26 | 27 | 27 | 27 | 27 |
| Number of ELC categories currently represented in the Spanish Network | 2 | 13 | 7 | 11 | 6 |
| Percentage of ELC categories currently represented in the Spanish Network | 8 | 48 | 26 | 41 | 22 |
| Number of ELC categories in the spatial gaps | 9 | 26 | 26 | 22 | 18 |
| Number of ELC categories in the priority ecogeographical gaps | 7 | 12 | 19 | 11 | 12 |
| Percentage of improvement in ecogeographical representativeness | 27 | 44 | 70 | 41 | 44 |

Table 5). The obtained percentage of ELC categories represented in the Spanish Network ranged from 8% to 48% in *Ae. biuncialis* and *Ae. geniculata*, respectively. As the priority ecogeographical gaps belong to categories not yet represented, their collection and conservation would contribute to increasing the percentage of ELC categories represented in the Spanish Network to values ranging from 27% to 70% in *Ae. biuncialis* and *Ae. neglecta*, respectively.

## Selection of priority collecting sites for traits of tolerance to drought and salinity

Among the 393 populations identified as priority ecogeographical gaps, 223 populations inhabit sites with a Lang index value <40, and thus are potentially adapted to arid environments (Table 6). The geographic location of these accessions is shown in Fig. 5A. The 20% of these 223 populations with the highest values of topsoil salinity for each species and their geographical distribution are shown in Table 6 and Fig. 5B, respectively. These 45 populations (Table 6) constitute the predictive characterization (PC) subset of *Aegilops* populations of potential interest due to their potential tolerance to drought and salinity, in addition to representing ecogeographical gaps. Table 7 contains complete information on the geographic location, the Lang aridity index and topsoil salinity of the populations included in this PC subset.

The inclusion of the predictive characterization subset populations in the genebanks of the Spanish Network on PGRFA would improve their ecogeographical representativeness between 7% and 11% (Table 8).

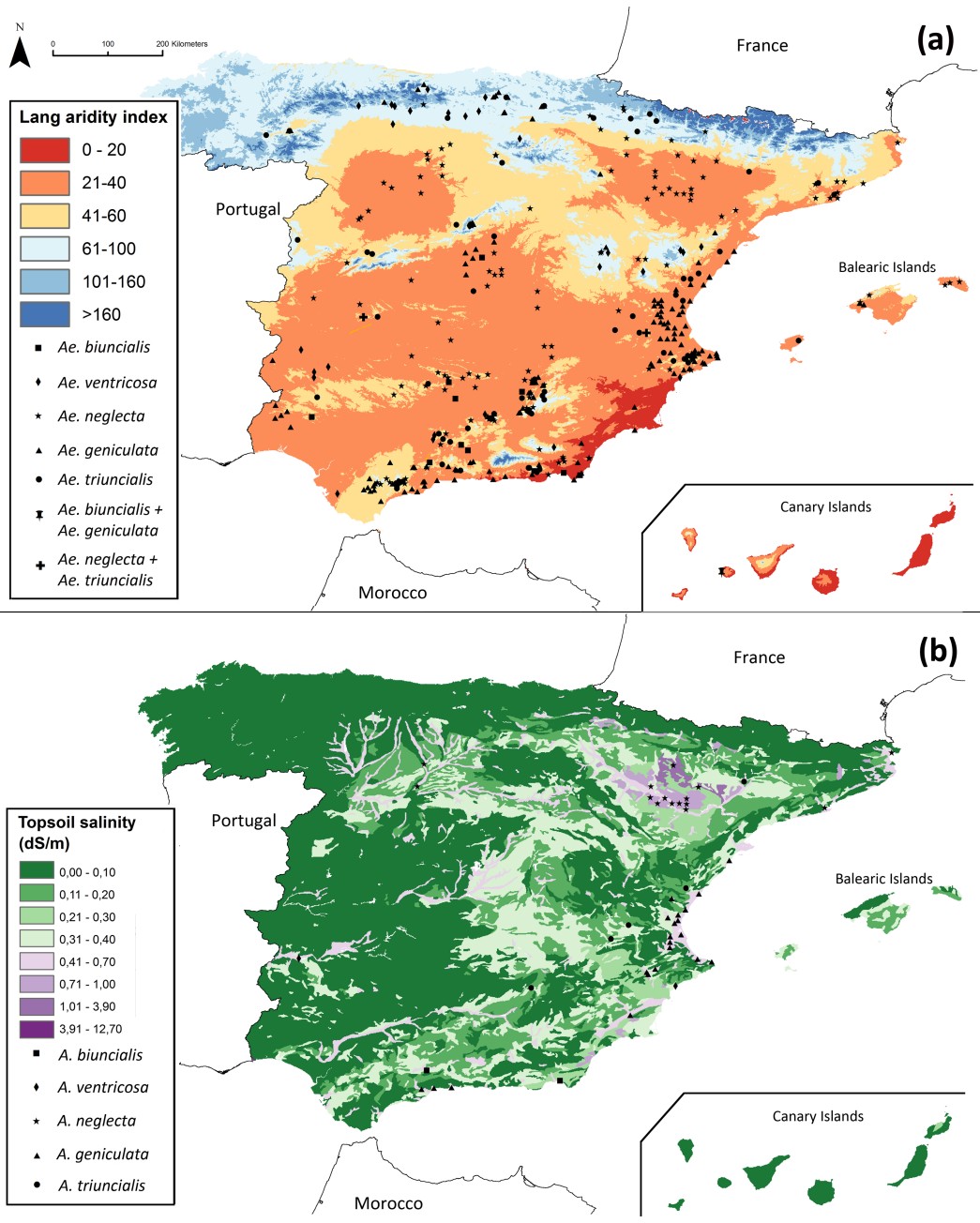

**Figure 5** Location of (A) priority ecogeographical gaps (PEG) of *Aegilops* in Spain that occur in sites where AI$_L$ < 40, and (B) PEG that occur in sites where the highest salinity values are also found.

## DISCUSSION

### The preprocessing of presence data

The quality of geo-referencing in spatial analysis is very important to obtain reliable results. *Maldonado et al. (2015)*, who studied the role of natural history collections in unveiling correct patterns of biodiversity and distribution, concluded that geographic inaccuracy affects diversity patterns more than taxonomic uncertainties. Similarly,

**Table 6  Priority ecogeographical gaps of targeted *Aegilops* species selected for drought and salinity tolerance in Spain.**

|  | *Ae. biuncialis* | *Ae. geniculata* | *Ae. neglecta* | *Ae. triuncialis* | *Ae. ventricosa* | TOTAL |
|---|---|---|---|---|---|---|
| Priority ecogeographic gaps with a Lang aridity index <40 | 10 | 103 | 76 | 26 | 8 | 223 |
| Priority ecogeographic gaps with a Lang aridity index <40 and with the highest topsoil salinity values | 2 | 21 | 15 | 5 | 2 | 45 |

*Graham et al. (2008)* evaluated how uncertainty in geo-references and associated location errors in occurrences influence species distribution modeling and found that models run with data subject to random location errors resulted in less accurate models in many species. However, few studies on genetic diversity or taxonomic spatial distribution describe robust methods to ensure the quality of geo-referenced data. Some authors, such as *Fielder et al. (2015)* and *Fielder et al. (2016)*, excluded records dated from before 1970, records lacking both coordinates and location descriptions and records with a precision lower than 4 km$^2$. *Ramírez-Villegas et al. (2010)* carried out a process to verify and correct the coordinates using BioGeomancer, Google Earth and highly detailed maps. *Khoury et al. (2015)* cross-checked the coordinates to country and verified that they occurred on land. After that, occurrence data were evaluated for correctness with experts on the target species. In our study, apart from removing accessions and external sources with low accuracy according to the established criteria, the assessment of the quality of the georeferenced data allowed us to identify the records with the highest quality. In this sense, the final number of records included in the analysis (considering both accessions and external sources) ranged between 30% and 53% of the initial number of records (for *Ae. geniculata* and *Ae. biuncialis*, respectively). This selective use of records, which complies with the minimum standards of georeferencing quality, reduces the probability of generating erroneous results in the analysis.

## Identification of spatial and ecogeographical gaps

Spatial bias in collecting activities and chorological studies often affects the spatial distribution of the species, as shown in *Maldonado et al. (2015)*. In our study, the spatial distribution of the accessions could reflect the intensity of collecting activities rather than the real distribution of the species (see Fig. 2A). A recent project focused on collecting *Aegilops* germplasm may have contributed to the difference observed in the numbers of accessions between areas because it was focused on the collection of *Ae. geniculata*, *Ae. neglecta, Ae. triuncialis* and *Ae. ventricosa* in the west, center and south of the country.

In a similar way, the distribution of the external sources of the targeted species (Fig. 2B) shows a higher number of populations in some areas of the country, such as the autonomous communities of Navarra and Valencia. This may reflect a higher intensity of chorological studies rather than a higher presence of the species in these areas. The
**Table 7 Geographic description of the Spanish *Aegilops* populations selected as potentially tolerant to drought and salinity.**

| Species | Latitude | Longitude | Province | Municipality | Lang index | Topsoil salinity (dS/m) |
|---|---|---|---|---|---|---|
| *Ae. neglecta* | 41.700833 | −0.045000 | Huesca | Villanueva de Sigena | 29 | 2.1 |
| *Ae. neglecta* | 42.060000 | −0.460000 | Huesca | NA | 36 | 2.1 |
| *Ae. neglecta* | 41.530000 | −0.840000 | Navarra | El Burgo de Ebro | 26 | 0.8 |
| *Ae. neglecta* | 41.430000 | −0.360000 | Zaragoza | Pina de Ebro | 27 | 0.8 |
| *Ae. neglecta* | 41.430000 | −0.720000 | Zaragoza | NA | 28 | 0.8 |
| *Ae. neglecta* | 41.420000 | −0.240000 | Zaragoza | Bujaraloz | 29 | 0.8 |
| *Ae. neglecta* | 41.510000 | −0.240000 | Zaragoza | La Almolda | 29 | 0.8 |
| *Ae. geniculata* | 38.640000 | −0.900000 | Alicante | Villena | 27 | 0.7 |
| *Ae. geniculata* | 36.736490 | −4.118400 | Málaga | Vélez-Málaga | 23 | 0.7 |
| *Ae. neglecta* | 41.430000 | −0.480000 | Zaragoza | NA | 23 | 0.7 |
| *Ae. neglecta* | 41.520000 | −0.600000 | Zaragoza | Osera de Ebro | 23 | 0.7 |
| *Ae. geniculata* | 39.137000 | −0.512550 | Valencia | Alberic | 24 | 0.7 |
| *Ae. geniculata* | 39.227020 | −0.509370 | Valencia | Alginet | 25 | 0.7 |
| *Ae. geniculata* | 39.936200 | −0.037130 | Castellón | Vila-real | 25 | 0.7 |
| *Ae. geniculata* | 39.450000 | −0.440000 | Valencia | Burjassot | 25 | 0.7 |
| *Ae. geniculata* | 39.494530 | −0.383580 | Valencia | Valencia | 25 | 0.7 |
| *Ae. geniculata* | 39.671850 | −0.260330 | Valencia | Sagunt | 26 | 0.7 |
| *Ae. geniculata* | 39.260000 | −0.330000 | Valencia | Valencia | 26 | 0.7 |
| *Ae. geniculata* | 38.603590 | −0.875550 | Alicante | Villena | 26 | 0.7 |
| *Ae. geniculata* | 39.857140 | −0.486740 | Castellón | Segorbe | 29 | 0.7 |
| *Ae. geniculata* | 38.856240 | −0.061490 | Alicante | Pego | 29 | 0.7 |
| *Ae. geniculata* | 40.487170 | 0.463370 | Castellón | Vinaròs | 30 | 0.7 |
| *Ae. geniculata* | 38.810000 | 0.173000 | Alicante | Jávea | 33 | 0.7 |
| *Ae. neglecta* | 41.710000 | −4.680000 | Valladolid | Cabezón de Pisuerga | 34 | 0.7 |
| *Ae. geniculata* | 36.706040 | −4.610470 | Málaga | Cártama | 34 | 0.7 |
| *Ae. neglecta* | 42.270278 | 3.144444 | Gerona | Roses | 36 | 0.7 |
| *Ae. neglecta* | 41.360000 | 2.040000 | Barcelona | Barcelona | 37 | 0.7 |
| *Ae. neglecta* | 42.080000 | −4.570000 | Palencia | Monzón de Campos | 37 | 0.7 |
| *Ae. biuncialis* | 37.021190 | −4.528030 | Málaga | Antequera | 39 | 0.7 |
| *Ae. geniculata* | 38.691500 | −0.757980 | Alicante | Beneixama | 30 | 0.7 |
| *Ae. geniculata* | 39.540000 | −0.550000 | Valencia | Bétera | 26 | 0.6 |
| *Ae. geniculata* | 37.930000 | −1.170000 | Murcia | Murcia | 17 | 0.6 |
| *Ae. ventricosa* | 38.869280 | −6.637140 | Badajoz | Lobón | 30 | 0.6 |
| *Ae. neglecta* | 41.330000 | −0.250000 | Zaragoza | Bujaraloz | 26 | 0.6 |
| *Ae. geniculata* | 39.046980 | −0.515710 | Valencia | Villanueva de Castellón | 25 | 0.6 |
| *Ae. neglecta* | 41.710000 | −0.830000 | Zaragoza | NA | 25 | 0.6 |
| *Ae. geniculata* | 39.584530 | −0.380200 | Valencia | Valencia | 26 | 0.6 |
| *Ae. geniculata* | 36.727000 | −4.405000 | Málaga | Málaga | 28 | 0.5 |
| *Ae. ventricosa* | 38.414300 | −0.423030 | Alicante | El Campello | 20 | 0.4 |

**Table 7** (*continued*)

| Species | Latitude | Longitude | Province | Municipality | Lang index | Topsoil salinity (dS/m) |
|---|---|---|---|---|---|---|
| *Ae. triuncialis* | 41.790000 | 0.710000 | Lérida | Castellón de Farfaña | 35 | 0.4 |
| *Ae. biuncialis* | 36.850000 | −2.330000 | Almería | Almería | 12 | 0.3 |
| *Ae. triuncialis* | 38.384490 | −2.804990 | Jaén | La Puerta de Segura | 27 | 0.3 |
| *Ae. triuncialis* | 40.030000 | −0.250000 | Castellón | Onda | 27 | 0.3 |
| *Ae. triuncialis* | 39.419940 | −1.199440 | Valencia | Requena | 28 | 0.3 |
| *Ae. triuncialis* | 39.190000 | −1.490000 | Albacete | Casas Ibáñez | 27 | 0.3 |

**Table 8** Number of ELC categories of *Aegilops* species currently represented in the Spanish Network and potential increase (%) in representativeness by collecting populations of the predictive characterization (PC) subset.

| | *Ae. biuncialis* | *Ae. geniculata* | *Ae. neglecta* | *Ae. triuncialis* | *Ae. ventricosa* |
|---|---|---|---|---|---|
| Number of categories in the ELC map | 26 | 27 | 27 | 27 | 27 |
| Number of ELC categories currently represented in the Spanish Network | 2 | 13 | 7 | 11 | 6 |
| Percentage of ELC categories currently represented in the Spanish Network | 8 | 48 | 26 | 41 | 22 |
| Number of populations in the PC subset | 2 | 21 | 15 | 5 | 2 |
| Number of ELC categories of the populations in the PC subset | 2 | 3 | 3 | 2 | 2 |
| Percentage of improvement in ecogeographical representativeness | 8 | 11 | 11 | 7 | 7 |

fact that the distribution of external sources is different from accessions indicates lack of collection missions in particular areas, for example the Canary Islands.

It is worthy to note that the real distribution of the species could be more extensive that the distribution shown by the combination of external sources and accessions. For instance, the target species might occur in places completely outside of the parameters used that are based on historical information. In this sense, it should be clearly understood that the efficacy of this collecting methodology, like any other, heavily depends on how well currently available chorological data represents the real distribution of the species.

Several chorological studies involving the spatial gap analysis of different species have been used as a guide for germplasm collecting. For instance, *Maxted et al. (2008)* identified *ex situ* conservation gaps in *Aegilops* germplasm collections as regions where the species were predicted to occur according to species distribution models, but previous collection had not taken place. In this study, the priority of germplasm collecting for each of *Aegilops* species was ranked high, medium or low, according to the number of germplasm accessions already conserved *ex situ* and the number of predicted under-sampled regions. *Shehadeh, Amri & Maxted (2013)* followed this methodology to carry out a gap analysis of *Lathyrus* L. species. Another recent methodology for gap analysis was described in *Ramírez-Villegas et al. (2010)* and applied to wild taxa of the *Phaseolus* genepool. It involves an eight-step process to evaluate conservation deficiencies at three different levels (taxonomic, geographic and environmental) through the calculation of sampling, geographic and environmental representativeness scores. This methodology was also applied in *Castañeda-Álvarez et al.*

*(2015)* to identify *ex situ* conservation priorities for the wild relatives of potato and in *Khoury et al. (2015)* in their study of the CWR pigeonpea.

The *ex situ* conservation gap analyses in these studies were obtained by overlapping the distribution maps of germplasm accession data and predictive distribution maps generated from the climatic envelope data of the accessions, or of both accessions and external sources.

With regard to identifying spatial gaps, *Ramírez-Villegas et al. (2010)* assessed the adequacy of geographic coverage of genebank accessions by means of a geographical representativeness score (GRS). This score is the geographic coverage of germplasm collections (modeled using the circular area statistic with a 50 km radius value) divided by the potential distribution coverage of the taxon under analysis. The higher the GRS is, the greater the representativeness of genebank collections is in relation to the potential distribution of the taxon. *Fielder et al. (2015)* and *Fielder et al. (2016)* also used this methodology to assess the geographical representativeness in their studies on the conservation of CWR in England and Scotland. They established a threshold of five accessions above which CWR are considered sufficiently represented in *ex situ* genebanks. In these geographic coverage assessments, the use of the potential distribution coverage of a taxon can lead to low GRS values due to the presence of false positives in the model (i.e., predicted locations with no real population occurrence) even though they have a good spatial representation in the *ex-situ* collections. In our study, we adopted a more conservative approach based only on recorded occurrences to avoid this problem.

With regard to identifying ecogeographical gaps, the exclusive use of species distribution models to develop strategies for germplasm collecting may also lead to the over-representation of some adaptive scenarios because these models guide collectors to the species' most preferred habitats (*Parra-Quijano, Iriondo & Torres, 2012b*).

On the other hand, predictive species distribution models based on bioclimatic information only offer a partial view of plant abiotic adaptation. In our study, we generated ecogeographical land characterization maps using the values of bioclimatic, geophysic and edaphic variables at species occurrence sites to suggest the potential adaptive scenarios of the species. The representativeness analysis, based on the comparison of the ELC categories of the accessions and the external sources considered spatial gaps, helped us to select priority collecting sites, avoid over-representation and identify populations from low and non-represented ecogeographical categories. This allowed us to identify populations in marginal environments in the species' range, where interesting traits related to abiotic stress tolerance may be found. The easy-to-use tools employed in this study (i.e., CAPFITOGEN tools, *Parra-Quijano et al., 2015*) allow genebank curators and technicians in charge of collecting activities to develop their own germplasm collecting design based on spatial and ecogeographical analyses.

A sound ecogeographical gap analysis requires that the resolution of the ecogeographical layers used is in a comparable scale to the resolution used for the chorological data. That is why, in our study, chorological data and ecogeographical data were filtered and/or chosen to accommodate to a similar resolution.

When trying to improve the ecogeographical representativeness of an *ex situ* germplasm collection, we would expect that the lower the number of accessions of a species in a

genebank is, the higher the probability of improving its ecogeographical representativeness by collecting seeds from new populations. However, this may not be the case. For instance, *Ae. biuncialis*, the target species with the lowest number of accessions in this study, is also the species with the lowest potential percentage of representativeness improvement. This is probably because the existing accessions are a good representation of the few ecogeographical categories where the presence of the species is recorded.

Including populations identified as priority ecogeographical gaps in the genebank collections of the Spanish Network on PGRFA would qualitatively improve ecogeographical representativeness, with increases in the percentages of ELC categories represented in the Network between 27% and 70%. Such increases are higher than those obtained in the *Lupinus* species collecting activities described in *Parra-Quijano, Iriondo & Torres (2012b)*, which ranged between 7% and 11%. However, it should be noted that the representativeness of *Lupinus* species, in the reference germplasm collection before the optimized collecting activities were carried out, was higher than in the case of the targeted *Aegilops* species.

## Selection of priority collecting sites for traits of tolerance to drought and salinity

The search for the 20% of the populations adapted to arid environments and with the highest topsoil salinity values allowed us to identify populations that occur in sites in the western Mediterranean distribution of the targeted species with the highest salinity values, according to the work on saline and sodic soils in the European Union by *Tóth et al. (2008)*.

One of the results of this study is the identification of 45 *Aegilops* populations of high interest due to their potential tolerance to drought and salinity, in addition to being ecogeographical gaps in the existing Spanish germplasm collections. Although predictive characterization is an inexpensive and effective approach to maximizing the likelihood of capturing a desirable level of trait expression among accessions of landraces and traditional varieties, this is the first time that predictive characterization has been applied to identifying wheat wild relatives with potential tolerance to drought and salinity. Interestingly, the validity of predictive characterization is presumably greater when used with wild relatives rather than when used with landraces, because wild relatives are more greatly affected by natural selection (no artificial selection). Thus, the link between existing environmental conditions and genotypes with local genetic adaptation is likely to be stronger than the link between environmental conditions and landraces or modern varieties, which is explored by FIGS.

In spite of having different geographic distributions, all studied *Aegilops* species occur in a variety of soil bedrock types (although predominantly in limestone), have a considerable precipitation range and are largely autogamous. The latter implies that genetic diversity is likely to be small within populations while local adaptation can generate large genetic differentiation among populations subject to different environmental conditions. Priority ecogeographical gaps of *Aegilops* that occur in sites where the Lang aridity index is <40 (Fig. 5A) show a good representation of all five species, including *Ae. neglecta* and *Ae. triuncialis*, which can also been found in habitats more humid than those found in the other three species. The selection for the highest topsoil salinity values (Fig. 5B) generated

a more heterogenous representation of the five species. Although *Ae. ventricosa* is the only species explicitly characterized in saline locations, just two populations of this species where selected in the PC subset for salinity tolerance. On the other hand, *Ae. geniculata* and *Ae. neglecta*, were the most represented in this subset (47 and 33%, respectively). This is largely explained by their large distribution area and great number of recorded populations. This implies that both species with more narrowly specialized habitats and those with larger distribution and wider ecological range can positively contribute genetic adaptations to particular environmental conditions.

Until now, the search for tolerance to drought and salinity in wheat has been carried out using approaches such as those in *Colmer, Flowers & Munns (2006)*, *Mólnar et al. (2004)*, *Farroq (2002)* and *Xing et al. (1993)*. FIGS approaches in cultivated wheat have focused on resistance to biotic factors such as plagues and diseases (*Endresen, 2011*; *Endresen et al., 2012*; *El Bouhssini et al., 2009*; *El Bouhssini et al., 2011*; *Bhullar et al., 2009*). Nevertheless, in other crops FIGS has been successfully applied in the search for abiotic stress tolerance. For example, *Khazaei et al. (2013)* tested the effectiveness of FIGS to search for traits related to drought adaptation in a large faba bean (*Vicia faba* L.) collection.

According to predictive characterization methods (*Thormann et al., 2016*), the probability of capturing phenotypes tolerant to drought and salinity in the predictive characterization subset would be higher than in a randomly chosen set. The 45 priority populations identified by the predictive characterization approach in this study are now considered priority collection populations for further optimized collecting activities of *Aegilops* germplasm. They will be shortly looked for and collected for *ex situ* conservation. When planning the collecting mission, the combined skills of a good field botanist, genetic resources expert and ecologist taking part in the collecting team should be used to critically assess the predictions obtained with this analysis and make further predictions for sites with potentially salt and drought tolerant germplasm.

Quality limitations of currently available species distribution and ecogeographic data may generate priority collection sites where the species no longer exist or the estimated ecogeographic conditions do not actually take place. Therefore, it is essential that the aridity and topsoil salinity conditions are verified at the site at the same time seeds are collected in the population. Then, the collected populations will be assessed for their tolerance to drought and salinity in order to validate the methodology.

## CONCLUSIONS

This study aimed to identify spatial and ecogeographical gaps in the Spanish germplasm collection of *Aegilops* and priority populations of potential interest due to their possible tolerance to drought and salinity. The methodology employed allowed us to establish an optimized collecting strategy by filtering potential collecting sites, thereby avoiding over-representation and identifying populations from low and non-represented ecogeographical categories. It also provided a subset of 45 populations of potential interest in terms of tolerance to drought and salinity. Because current knowledge of the real distribution of this species is still incomplete, this methodological approach should be observed in dynamic

terms. As the quantity and quality of species distribution data improves, this type of study should be carried out again and the collection priorities should be updated accordingly.

Subsequent collections and evaluations of these populations will provide essential feedback on the efficacy of these approaches to improving the genetic representativity of genebank collections and identify genotypes with desired traits.

## ACKNOWLEDGEMENTS

We would like to thank the personnel at CRF-INIA. We are also grateful to Lori De Hond for her linguistic assistance.

### Funding

The authors received no funding for this work.

### Competing Interests

The authors declare there are no competing interests.

### Author Contributions

- Rosa María Garcia conceived and designed the experiments, performed the experiments, analyzed the data, contributed reagents/materials/analysis tools, wrote the paper, prepared figures and/or tables.
- Mauricio Parra-Quijano and Jose María Iriondo conceived and designed the experiments, analyzed the data, contributed reagents/materials/analysis tools, reviewed drafts of the paper.

### Data Availability

The raw data has been supplied as Supplementary Files.

### Supplemental Information

Supplemental information for this article can be found online at http://dx.doi.org/10.7717/peerj.3494#supplemental-information.

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
