# Peer review of "Identification of ecogeographical gaps in the Spanish Aegilops collections with potential tolerance to drought and salinity"

_PeerJ, doi:10.7717/peerj.3494_

## Round 0.1 · original submission · Major Revisions

Thank you for your recent manuscript submission. There is a great need for identifying diverse germplasm for utility of developing new traits in plant breeding programs. If I were reading this manuscript as a proposal to search for more diverse germplasm I would highly align with supporting the effort; however, I think that the general comments reflect some additional effort or evidence needs to be provided to move this work from a predicted-state to a validated-state.

There are also a wide range of comments referring to the clarity of the figures with perhaps too many features illustrated. For the density of data provided perhaps it would require an adjustment to what is actually needed to illustrate the points in the map figures. Along with some other issues with typographical errors and readability, the manuscript would require more scrutiny toward additional clarity. I did think the manuscript was reference-rich and helps define the importance for seeking germplasm diversity.

I will consider this manuscript as requiring major revision, and with attention to the comments made by the reviewers. All reviewers presented important concerns which will need to be addressed. I look forward to an amended version of the manuscript.

·

Basic reporting

It is an interesting paper on the collection strategies and the importance of ecogeography indications to perform proper collections and fill the gaps present in the collections. Unfortunately reviewing it was highly time consuming since the analyses performed and the choice of data was not clear and I disagree with some of their data and results interpretation. I reported most of the corrections (also typographical) and comments within the paper itself (file attached). Nevertheless, some others are herein reported.

The two paper objects reported at the end of Introduction, seems to me a bit repetitively. In synthesis they want identify 1) identify uncollected ecogeographical areas to be collected, and 2) identify salt/drought tolerance un-collected populations to be collected. Actually point 1 seems to me to include also point 2.
Actually I would change the second into 2) to identify populations (collected or to be collected) that might be of potential interest because of their tolerance to drought and salinity.

Please consider that abbreviation of Aegilops is Ae. and not A. change all across the paper.

Experimental design

I don’t understand why they removed accessions with geographic coordinates in sexagesimal rather than convert them in the decimal system. Please clarify.

The bioclimatic variables are 17, as reported in Table S1 not 37, as in the text. Please correct also Fig. 1.

About the statements on rows 29-37 of page 8 regarding the not homogeneous distribution of accessions in the studied area, it should be mentioned (supported with evidences, probably in the introduction or discussion rather than in the results) if this is due of the lack of mission in some areas or to the lack of species presence. The fact that the distribution of external sources is different from the one of internal sources indicate that the lack/differences is simply due to the lack of collection missions in particular area. For instance, probably in some of the Canary Islands nobody run any collection. Since it is reported then in the discussion rows 6-9 page 12, move all there.

Row 7-14 of page 9 seems Material and Methods rather than Results. Please move it.

Please report in Fig 3 that the ecogeographical categories are described in Table S2 and/or Table 2.

I think that the creation of several ecogeographical maps one for species create confusion rather than increasing information. Parra-Quijano et al. (2012a) reported a single ELC map for Spain; in fact the Ecogeographical land characterization seems to be related to “Ecogeographical” characteristics of the land and not to the species. So I would aspect a single map for all the Spain also here and not different maps for the different species.
If some ecogeographical characteristics come out to be more important than others (as reported on Table 2) for some species, it is fine, but would be clearer to had in Fig 3 the general unique ecogeographical map for all Spain. In any event, it is not clear why if in Table 2 are selected different variables in each Aegilops species and ecogeographical component according to the importance function of the random forest approach, but then on Table S2 are reported, for each ELC category, the values of the same 8 variables for all the species. Please clarify.
Is it possible to underlining if there are ecogeographical categories common among species? Would be more interesting to have a single general list of ecogeographical categories, rather than several lists one for each species.
As reported by the authors, their description of the ecogeographical categories distribution have “similarities among the five maps”, but this is just an opinion based on feelings rather than statistic evidence, again it would be nice to have an unique list of ecogeographical categories for all the species.

Validity of the findings

On Table S2 for A. ventricosa the “Temperature seasonality” colum reports values which seems wrong. Please check them.

It is not clear how Fig 4 is obtained. For example since from Fig 2 there are not sample from Galicia, and from Fig 3 Galicia has particular ELC, I would expected in Fig 4 Galicia has a land to be sample since has high collection gaps.
To a less extend this is also true for Murcia and Canary Island (Fuerteventura and Lanzarote), particularly looking to Figs 2, 3, 4 and 5 and considering that no samples are from there. Please explain.

Please control Table 6, as far as I know Valencia is not a Municipality of Murcia province, and Murcia is not a Municipality of Álava.

Additional comments

I think the paper would be strongly improved by simplify it reducing the ecogeograpical area to a single one and reporting the gaps as point where the collection is missing, and not in a not well described points collected by external sources.

·

Basic reporting

No Comment. Excellent.

Experimental design

No Comment, except as noted in next section.

Validity of the findings

The authors have done an excellent job in locating and analysing the data on ecogeographic distributions of 5 Aegilops species. They have concluded that gaps in the genebank collections occur. That is fine. Then they state that populations have been identified that are targets for future collections. I don't think that is what is meant or found in the study because they have not verified . The places where they think Ae species exist should be visited. Ground-truthing is essential. This is important work, but it falls short on that critical issue, i.e. see the populations and collect seed from them as multiplant collections to represent the populations in the field. Remember the experience of Charles Rick in collecting tomato wild relatives in South America. Places where collect no longer had tomato species when he went back 2 decades later. The goats had wiped them out. Gap analysis can be very useful, and this study is a good example. One gap in the process is that the target species might occur in places completely outside of parameters used that are based on historical information..

Additional comments

Please get in a Land Rover and visit the various sites that have been identified as potential extant homes for endemic Aegilops species that might have been seen, but not collected for ex situ conservation.. You then can complete the study according to the scientific method. I really like this work and the attention to details of studying prior collections and site recordings. I can agree that this paper is publishable, but you should be clear about next steps needed validate the data. That can be subject of the next paper.

Reviewer 3 ·

Basic reporting

Article uses English in a clear way, some small typos are noted in the General Comments section, below. Introduction and background are adequate. Since the main premise of the article is that something can be learned about the environmental range of collected material for a species with respect to the environmental range of the species as a whole, it is surprising that no review of what is known about the autoecology, distribution, and breeding systems of the target species. Overall, the sense of the article is that it is a database manipulation exercise divorced from any attempt to relate the results with any information from any of the actual accession sites. Three of the figures (2, 3, and 4), while relevant, include so much data, lumping together all target species, that the reader cannot easily grasp the message. See specific comments in General Comments section below. References are not completely in alphabetical order by author: for example, note the placement of Jarvis et al. 2008 and Hijmans et al. 2005.

Experimental design

For the database manipulations, comparing and associating accession information such as source location with independently derived spatial and potentially ecogeographic information, the study has been conducted rigorously and with adequate information to be reproducible. However, the connections of the parameters used to the actual growing conditions of the accessions is hypothetical and there was no attempt to even use some accessions from a single target species as a case study, visiting the site and evaluating the correspondence of conditions on the ground with that predicted by the sole environmental parameters: aridity and topsoil salinity. Further the five target species are treated as if they were all the same. Surely there are indicators from their biology, breeding behavior, and phenology that could inform the analyses.

Validity of the findings

The conclusion of the study is essentially a hypothesis whose validation is left to future analysis. Figure 1 indicates a total of 57 bioclimatic, edaphic, and geophysical variables were considered in the process of selection of candidate accessions. However, the relationship of the scale at which these variables were derived with the scale of the source locations of the accessions is not evaluated or otherwise discussed. Table S1 defines these variables, but the source reference for them are three global databases/surveys. How can this scale be meaningfully reconciled to the scale of the Spanish collections? For the ELC categories of Table S2, I have a similar question: How does the scale of these values match the scale of the collection sites? The provided Spanish inventory database of accessions of the targeted species does not give any confidence of an ability to usefully match these vaguely defined sites to global aridity and salinity databases. One of the two external databases (Raw_data_GBIF) add lat/long information for accessions, but again the specificity of the site location of an accessions does not seem to match the scale of the bioclimatic/edaphic/geophysical variables available. Finally, I am bothered by all the accessions excluded from further study because of insufficient georeferencing which is obviously a criterion for the type of analysis conducted here. Could the combined skills of a good field botanist, genetic resources expert, and ecologist use an understanding of a species and the habitats where it is found along with floristic data to assess gaps in the national collections and make predictions for sites with potentially salt and drought tolerant germplasm?

Additional comments

Line: note/comment/question
55 missing word: ‘45% ^under^ drought’
87 ‘occur’ should be plural: ‘occurs’
119-120 This single-sentence paragraph should be appended to start of next paragraph.
129-131 Presumably the resistance to sunn pest was in wheat, like that for RWA, but sentence needs rewriting to make this clear.
172-173 + GRIN’s use of taxonomy is not authoritative, so why use GRIN taxonomy as authority when an exception must be made anyway? Better reference would be van Slageren (1994), which is a taxonomic authority and whose conclusions and usage conforms to what is used in this paper.
van Slageren M.W. 1994. Wild wheats: A monograph of Aegilops L. and Amblyopyrum (Jaub. & Spach) Eig. Wageningen Agricultural University, Veenman Drukkers, Wageningen, The Netherlands.
178-184 This is a very questionable step biologically, but obviously required because of the limitations of the bioinformatics approach taken. It would be useful to tell the reader the overall count of these excluded accessions, without having the reader need to do his/her own calculation from Table 1.
224-230 In spite of prefacing the word gap with the adjectives spatial and ecological, in reality, these gaps are entirely constructs of the database manipulations and have not been ground-truthed to be actual spatial or especially ecological gaps.
245-248 This is the only attempt to relate the results of the database manipulations to reality on the ground at the source sites of accessions, and it seems lacking. There is no discussion of the relative granularity of such metrics as the aridity index and the topsoil salinity values. Do the passport data for an accession include the topsoil salinity at the site at time of collection? Are there measures of annual precipitation and temperatures available for each collection site at time of collection? Just the name of the metric ‘topsoil salinity’ suggests it was measured in an agricultural context. Were coastal marsh areas included in the topsoil salinity assessment database?
369 tense: ‘ran’ should be ‘run’
380-384 Two things here: Shouldn’t there have been some preliminary evaluation of the overall proportion of ‘acceptably’ georeferenced data to the total amount of accessions available in national and international genebanks for each species? Second, I don’t agree that this ‘selective use of records’ guarantees accurate analyses. Unless there is further analysis on a species-by-species basis for how representative of the distribution with respect to habitat and environment the selected subset of records are, these analyses have no useful predictive power beyond the explicit set of records.
398 missing space: were ^space^ predicted
465-467 The relationship of the identified ‘45 Aegilops populations’ and them being potentially adapted to arid environments and topsoil salinity is very tenuous, almost a leap of faith, given the absence of even a case study of one or two accessions per species with a visit to the actual site of their collection.
489-491 Only here at the very end is it mentioned that real-world evaluation of drought and salinity tolerance needs to be established before this database manipulation strategy can be construed to be useful.
Figure 2 Putting all accessions of each species together in these two maps completely overwhelms the ability of the reader to appreciate just what the species-specific situation is for each species between the in-country collections and those held externally. Better to do this for one species as an example and put the rest in the supplemental material.
Figure 3 The individual maps are too small and the number of ELC categories are too many to allow the reader to understand these maps. Why doesn’t the legend give some indication of the meaning or value of a category? The accompanying text where the figure is cited is of no help either.
Figure 4 legend missing space: as ^space^ priority. Again, lumping all five species into the single map makes it difficult to understand the import of this information.
Table 1 Why not also provide a proportion of the excluded accessions at each step from the total started with?
Table 2 It would be very useful to have some discussion about the ecology of each species and why these particular ‘important’ components that emerged from the analysis might even be relevant.
Table 4 Why does Ae. biuncialis have one fewer ELC category?
Table 6 What is the relationship of actual sites at which the two metrics (aridity and salinity) were taken and the sites at which the accessions were collected? Is the lat/long specific to the accession collection site or to the municipality?
Table S1

Was any attempt made to confirm that a genebank record actually means seed available?

It is telling that the flow chart of the database manipulations in Fig. 1 does not have any element of ground truthing, such as actually visiting any site and evaluating the match of the environmental variables to the site.

I am bothered by all the accessions excluded from further study because of insufficient georeferencing which is obviously a criterion for the type of analysis conducted here. Could the combined skills of a good field botanist, genetic resources expert, and ecologist use an understanding of a species and the habitats where it is found along with floristic data to assess gaps in the national collections and make predictions for sites with potentially salt and drought tolerant germplasm?

---

## Round 0.2 · Minor Revisions

The latest version of your manuscript reads very well and requires little, or no edits. The efforts produced here result to be at least a specialized bioinformatic procedure to mine available information on Aegilops germplasm. As mentioned in one of the earlier reviews a big step in adding validation to this study would be to actually go out and identify the presence of the germplam in the proposed sites as an attempt at closure. As this suggestion was not done it would be important for any follow-up work to refer to this effort as being successful in guidance toward obtaining trait associated germplasm; unless this is done, this work will amount into a carefully generated hypothesis that will simply serve as a road-map to an unpromised destination. The evidence provided does show promise; however, it is more less an elegant process of setting out a plan for future work. As this manuscript may have been thoroughly reviewed and the suggestions made from reputable sources a recommendation for publication is submitted; the readership and feedback from this manuscript will stand as a test of time for its utility until the proposed hypotheses can be ultimately validated. It would be appreciated if you can address some of the voiced concerns in a remaining, final Minor Revision; otherwise, it will be accepted as such. There is a need for these analyses to help develop crop resources for the future; congratulations on your efforts.

·

Basic reporting

The authors accept most of my and other reviewers suggestions with a consequent improvement of the manuscript. Nevertheless, there are something which still needs to be clarified.



Rows 253-255 the sentence “following parameters: 30-arc-second cell size and 8 clusters as a maximum number of cluster per ecogeographical component (bioclimatic, edaphic and geophysic), considering the area of Spain.” Is unclear.

Experimental design

The choice to use several ELC maps should be better described, not only to answer my points, but also in the manuscript. From the Introduction (rows 99-106) it seems logical a single map.
I still think it would be very interesting, and easier, to have a single map with the ecogeographical variables (which are the variables which describe a territory regardless the species presences) and then see how the different species react and are distributed in relation to that.

Validity of the findings

The study is interesting and their findings useful for further collection missions.

·

Basic reporting

I have previously reviewed this paper. Now I have seen two additional reviews and the responses of the authors. I still am concerned that ground-trothing has not been done to validate their 'gap' conclusions. I am pleased that the authors agree and have made some preparations to do the necessary field work. The assessment is thorough and relevant to Aegilopodists. Since this analysis meets rigorous standards, it does stand alone and provides the basis for their next paper which I hope will judge their predictions and also provide more ex situ genetic resources for researchers..

Experimental design

No comment

Validity of the findings

No comment

Additional comments

See my note at beginning of this review

Reviewer 3 ·

Basic reporting

No comment, see General Comments.

Experimental design

No comment, see General Comments.

Validity of the findings

No comment, see General Comments.

Additional comments

I think it is instructive that all three reviewers called attention to similar issues with the paper, several focusing on the absence of any real-world attempt to frame the study before embarking on all the data and metadata analyses.

The only major concession to the main points of the reviewers that I can see is the call for the validation by a well-appointed collecting team in the Discussion and Conclusion.

It’s clear that this author team is not equipped to undertake what would best inform this whole process and wants/needs to salvage a publication from their database manipulations. The question is what should the journal do with this mss. A danger of publishing it essentially as it now is (the issues below should be addressed) is that its conclusions become the standard for future collection efforts, preventing any worker from obtaining support to do what was needed before all this work was done: Take a couple of these species, review holdings (in Spain’s genebanks, if they must restrict it in that way), and visit some sites that by environment might suggest salt tolerance (marine salt marshes?) and from which ‘external’ collections had been made, and see if new collections from those sites can be made.

I think the superlative adjectives should be removed: i.e., ‘identify the most important ecogeographical variables’ should be ‘suggest ecogeographical variables that might be important’.

The text at lines 447 to 448: I am not convinced that the ‘different adaptive scenarios of the species’ were described as accurately as possible. Maybe ‘potential adaptive scenarios of the species were suggested’ and that’s all.

Terms like ‘accuracy’ are relevant within the context of the database analyses conducted, but should not be extrapolated to the interpretation of the analyses with respect to environments and species distributions.

Lines 528 to 530 capture the essence of what I’m trying to say. Until this is done, the approach in this paper should be presented as hypothetically useful, providing some hypotheses which could be tested, not presented as an end result of the identification of real and to-be-exploited gaps.

Accordingly if it’s going to be published as is, I recommend the title be changed to ‘Suggestions of ecogeographical…’

The following additional points should be dealt with before publication.

1. Reviewer #1 correctly pointed out that Aegilops should be abbreviated as Ae. and this has been done in text and tables, but was not done in the figures (e.g., Figure 5).

2. New Table 1: Adds summaries of biological information about each species. However, the information has no context in the paper other than to say ‘here they are’. The recommendation was that perhaps the autecological information could be reviewed with conclusions drawn about the likelihood of a species being found in drought or saline impacted environments. Nothing is gained by just inserting the table.

The legend of the table says that unless otherwise noted, all information comes from van Slageren (1994) so it makes no sense to redundantly cite van Slageren as in last column/row, Mating system for Ae. ventricosa.

3. While van Slageren (1994) is now cited in several places, the reference does not appear in the References section.

4. Figure 4. Now that this figure shows only a single species, Ae. geniculata, other questions emerge. If these marked sites are ‘priority’ for this species, it seems like now there are two levels of priority. One is that for this Figure and higher levels are indicated in Figure 5, which shows one priority based on aridity (a, AI<40) and one based on salinity (b, ‘highest’). What should I conclude from the fact that some sites marked as priority in Figure 4 are not marked in Figure 5 (either in a or b). For example, in Figure 4, there is a single Ae. geniculata site in the Canary Islands, but in Figure 5, the Canary Islands have no apparent Ae. geniculata sites. There is an Ae. biuncialis site marker. Is that a mistake for the Ae. geniculata marker? Or are both markers really there, but the resolution of the graphic doesn’t allow the distinction? If the latter, that’s another argument for not putting all species in a single figure.

5. While I think it is useful that a single species is used in the text figures, as an example, putting the figures with the other species into the supplemental information, I expected some discussion of Ae. geniculata in the text. Given the priority sites you show for it (Figures 4 and 5), is there any meaning in these sites with respect to what is known about the species’ autoecology and the general environment of those sites with regard to the goals of drought resistance and salinity tolerance?

---

## Round 0.3 · accepted · Accept

There were many concerns with this manuscript at first and several concessions had to be made; however, with all the back and forth suggestions and revisions this is a significant effort which may lay the foundation for future work. The reviewers were very helpful in this respect. I applaud you on your efforts and can anticipate that the readership will appreciate the work as well. Consider this manuscript accepted for publication. Congratulations.